# Effects of Harvest Timing on Phytochemical Composition in *Lamiaceae* Plants under an Environment-Controlled System

**DOI:** 10.3390/antiox12111909

**Published:** 2023-10-25

**Authors:** Da-Hye Ryu, Jwa-Yeong Cho, Seung-Hoon Yang, Ho-Youn Kim

**Affiliations:** 1Smart Farm Research Center, Korea Institute of Science and Technology (KIST), Gangneung 25451, Republic of Korea; dahyeryu0507@hanmail.net (D.-H.R.); chocho7023@naver.com (J.-Y.C.); 2Division of Bio-Medical Science and Technology, KIST School, Korea University of Science and Technology (UST), Daejeon 34113, Republic of Korea; 3Department of Biomedical Engineering, College of Life Science and Biotechnology, Dongguk University, Seoul 04620, Republic of Korea; shyang@dongguk.edu

**Keywords:** *Lamiaceae* plants, harvest time, phytochemicals, rosmarinic acid, VOCs, antioxidant activities

## Abstract

The *Lamiaceae* family is widely recognized for its production of essential oils and phenolic compounds that have promising value as pharmaceutical materials. However, the impact of environmental conditions and different harvest stages on the phytochemical composition of *Lamiaceae* plants remains poorly understood. This study aimed to investigate the effects of harvest time on the phytochemical composition, including rosmarinic acid (RA) and volatile organic compounds (VOCs), of four *Lamiaceae* plants—Korean mint (AR), lemon balm (MO), opal basil (OBP), and sage (SO)—and was conducted under an environment-controlled system. Although all four plants had RA as the dominant compound, its distribution varied by species. The flowered plants, including AR and OBP, exhibited a rapid increase of RA during the transition from the vegetative stage to the reproductive stage. In contrast, non-flowered groups, including MO and SO, showed a steady increase in the content of total phenolics and RA. The main components of VOCs also differed depending on the plant, with characteristic fragrance compounds identified for each one (AR: estragole; MO: (Z)-neral and geranial; OBP: methyl eugenol, eugenol, and linalool; and SO: (Z)-thujone, camphor, and humulene). The total VOCs content was highest on the 60th day after transplanting regardless of the species, while the trends of total phenolics, RA content, and antioxidant activities were different depending on whether plant species flowered during the cultivation cycle. There was a steady increase in species that had not flowered, and the highest content and activity of the flowering period were confirmed in the flowering plant species.

## 1. Introduction

Numerous herbs belong to the family *Lamiaceae* [1], which has a broad industrial presence across various fields. In particular, their culinary use has recently extended to include applications in cosmeceuticals and cosmetic–pharmaceutical hybrid products [2]. Various phenolics, including flavonoids, as well as their characteristic fragrance, have been found to contribute to their antimicrobial and antioxidant properties [3,4,5]. Along with these trends, the increasing demand for natural products has seen herbs considered as promising crops owing to the preference for the use of natural materials to replace synthetic compounds [6]. The major pharmacologically active compositions of herbs, rosmarinic acid (RA), and volatile organic compounds (VOCs) have already been identified [7]. In particular, RA is the main phenolic component of herbs [8,9]. RA is a naturally occurring hydroxycinnamic acid ester (caffeoyl derivative) and has been revealed to possess biological potential, including antibacterial, antiviral, anti-inflammatory, anticancer, and anti-angiogenic activities [10,11]. Moreover, RA is reported to have important properties for industrial applications, including its ability to induce melanogenesis for photoprotection against UV-A and UV-B absorption, as well as its antistaphylococcus aureus and anti-inflammatory activities, which make it suitable as an anti-acne agent [12,13]. The strong antioxidant activity of RA, which has been proven to be the most prevalent phenolic compound in sweet basil, thyme, marjoram, sage, rosemary, and lemon balm, is associated with these herbs [14]. Phenolics, including flavonoids, possess powerful antioxidants properties, as they can effectively scavenge free radicals, singlet oxygen, and superoxide radicals due to their hydroxyl groups, and interact with enzyme functions [15]. In various plants, the content of rosmarinic acid (RAC) influences antioxidant properties and a strong correlation has been observed between phenolic content and antioxidant activities in species like *Perilla frutescens* [16] and *Melissa officinalis* [17]. On the other hand, VOCs are another characteristic feature, and their emission exhibits considerable variation, including monoterpenoids and sesquiterpenoids [18]. In these classes, the predominant VOCs include eugenol, methyl eugenol, linalool, camphor, citral (neral and geranial), estragole, β-pinene, and β-caryophyllene [19]. Additionally, VOCs are considered promising metabolites that are known for their safety and utilized for their antioxidant, antibacterial, and antimutagenic activities [20]. RA and VOCs can fluctuate wildly by not only the genetic variability, plant parts, and environmental conditions, but also different developmental stages [21,22,23,24].

The synthesis and accumulation of secondary metabolites in plants show a species-specific pattern, and the expression of genes according to the cultivation stage effects. During the different growth stages of plants, a multitude of enzymes play a role in the pathways responsible for phenolic biosynthesis [25]. Consequently, research has explored the influence of growth stages on thyme [26,27], oregano [28,29], and rosemary [23,24], revealing insights into their metabolites and related biological activities, including antimicrobial [30] and antioxidant properties [31]. Environmental conditions also play a significant role. Factors such as plant physiology, metabolism, and cultivation conditions can lead to alterations in the expression of numerous enzymes involved in phenolic biosynthesis pathways, thus influencing interactions with plant secondary metabolites [25]. These enzymes are governed by intricate regulatory mechanisms and exhibit precise responses to environmental stimuli, including those associated with seasonal changes [25].

Several specifications were selected by referring to previous studies. Shekarchi et al. [32] conducted a comparative experiment on the content of rosemary acids in a total of 29 specifications, and through this, a high content of RA contained in lemon balm (*Melissa officinalis,* MO) and sage (*Salvia officinalis*, SO) was confirmed, which were similarly reported in other papers [33,34]. Furthermore, *Agastache rugosa* (AR) and *Ocimum basilicum* Purpurascens (OBP), which have been used in pharmaceutical benefits and are commonly consumed as foods, were investigated in this study [35,36]. To achieve commercially consistent high-quality and high-functionality production of *Lamiaceae* plants, cultivation patterns, functional components, and antioxidant activity were evaluated based on the cultivation period under an environment-controlled system.

## 2. Materials and Methods

### 2.1. Cultivation of Plants

AR, MO, OBP, and SO seeds were purchased from Aram Seed Co. (Seoul, Republic of Korea), N. L. Chrestensen Erfurt Seeds (Erfurt, Germany), Danong Seed Co. (Dongducheon-si, Gyeonggi-do, Republic of Korea), and Worldseed Co. (Gwangju-si, Gyeonggi-do, Republic of Korea), respectively. Seeds of the selected plants were sown into the soil (volume density, 0.3 mg/m^3^; pH, 5–7; EC, ≤1.2 ds/m) and grown at the vertical farming system of the Korea Institute of Science and Technology (Gangneung, Republic of Korea) at temperatures ranging from 18 to 23 °C and a day/night period of 14/10 h (Appendix A).

One-month-old seedlings were transplanted into trays packed and cultured in a greenhouse, which included actuators (shield, cooler, and heater) and a monitoring system. Climate data during the experiment are provided in Table 1. Whole parts of plants collected at 30, 60, 70, and 80 days after sowing (DAS) were rinsed and weighed. 

The growth rate is calculated according to the changes in diverse factors such as dry weight, leave area, and height, and it can be used for plant growth analysis [37,38]. In our study, plant weight was subjected to growth rate parameters according to the following formula:$$\mathrm{Weight}\;\mathrm{rate}=\frac{W2-W1}{t2-t1}$$
where W1 and W2 are plant weights at times t1 and t2.

### 2.2. Sample Preparation

The whole plants were frozen in liquid nitrogen and stored at −80 °C for 1 day. The samples were dried in a freeze-drier (−80 °C) for 1 week and finely ground in a mortar. The finely ground samples were extracted with 70% ethanol (2 g per 40 mL) using a sonication extractor (Bandelin Sonorex, Bandelin Electronic, Berlin, Germany) at 60 °C for 2 h. Extracts were filtered through Whatman No. 2 filter paper and concentrated in a nitrogen evaporator (Allsheng MD 200, Hangzhou Allsheng Instrument Co., Ltd., Hangzhou, China). Dried crude extracts were redissolved in dimethyl sulfoxide and filtered through an analytical 0.22 μm polyvinylidene fluoride membrane filter before in vitro assays, qualitative analysis (UPLC-MS/MS), and quantitative analysis (HPLC).

### 2.3. Analysis of Total Phenols

TPC was measured using the Folin–Ciocalteu phenol method [39] with some modifications. Plant samples (10 μL each) were added to 2% Na_2_CO_3_ (200 μL) and mixed for 3 min in a 96-well plate. The mixture was reacted with 1 mol/L Folin–Ciocalteu phenol (10 μL) for 27 min at room temperature, after which absorbance was measured at 750 nm with a spectrophotometer (Bio-Tek Instruments, Winooski, VT, USA). The TPC was then calculated using a calibration curve; gallic acid was used as the standard, and the results were expressed as mg GAE (gallic acid equivalents) per g of dried weight.

### 2.4. Identification of Rosmarinic Acid by Ultra-Performance Liquid Chromatography with Tandem MS (UPLC-MS/MS)

UPLC-TQ-MS/MS was performed on an Agilent 1290 Infinity II LC system (Agilent, Waldbronn, Germany) coupled with an Agilent 6470 B Triple Quadrupole instrument. The injection volume was set as 1 μL and an injected sample was carried out at a column temperature of 45 °C on a YMC-Triart C18 column (100 × 2.0 mm l.D. S-1.9 μm, 8 nm). The mobile phase consisted of water containing 0.2% formic acid (A) and acetonitrile containing 0.2% formic acid (B). The flow rate of the mobile phase was set to 350 μL/min and a gradient system was applied for analysis. In the initial point, the B ratio was 10% and it was maintained for 1 min. The B ratio was increased to 40% at 8 min and 50% at 20 min. Finally, the B ratio was increased to 100% at 40 min and for column washing, there was a 2 min post-run. The detection wavelength was set at 280 nm and the detected peaks were proposed by TQ-MS/MS analysis. 

An Agilent 6470B Triple Quadrupole instrument was applied to MS and MS/MS detection. The operation conditions were set as follows: drying gas (N_2_) flow rate, 7 L/min; drying gas temperature, 325 °C; nebulizer, 25 psi; sheath gas flow rate, 7 L/min; sheath gas temperature, 250 °C; capillary, 3500 V; fragmentor, 135 V; collision energy, 25 V; scan mode, MS scan, and production mode; mass spectra recorded range, *m*/*z* 100–1000 at negative mode. Agilent Mass Hunter Workstation Acquisition Software Version B.05.01 and Qualitative Analysis Software Version B.07.00 were utilized for data acquisition and data processing. 

### 2.5. Quantitative Determination of Rosmarinic Acid

RAC was measured by high-performance liquid chromatography (Agilent 1200 series, Santa Clara, CA, USA) coupled to a diode array detector at 280 nm. A 10 μL sample was injected into the Supersil ODS-II (4.6 × 250 mm, particle size 5 μm, Dalian, China) and the column temperature was maintained at 40 °C. The flow rate was 1 mL/min and water containing 0.1% formic acid (A) and acetonitrile (B) were used as solvents. The following gradient solvent system was used: 0–1 min, 10% B; 1–15 min, 10–50% B; 15–25 min, 50–100% B; 25–26 min, 100–10% B, with a 2 min post-run period of 10% B. RA was identified by congruent retention times compared with standards and the content was calculated to be mg per g of dried weight.

### 2.6. DPPH Radical Scavenging Activity

DPPH radical scavenging activity was assessed to determine the antioxidant potential of the samples [40]. Briefly, DPPH solution was dissolved in ethanol to 0.15 mM, which was adjusted to an absorbance of 1.00 ± 0.05 at 517 nm. The sample (10 μL) was reacted with DPPH solution (190 μL) at room temperature for 30 min in the dark. After incubation, the absorbance was recorded at 517 nm with a UV/Vis spectrophotometer (Bio-Tek Instruments). DPPH radical scavenging activity (%) was presented according to the following formula:$$DPPH\;radical\;scavenging\;activity\left(\%\right)=\frac{{Abs}_{control}-{Abs}_{sample}}{{Abs}_{control}}\times 100$$
where Abs_sample_ is the absorbance of tested sample and Abs_control_ is the absorbance of the control after 30 min reaction.

### 2.7. ABTS Radical Scavenging Activity

ABTS free radical scavenging activity was determined using a previously described method with some modifications [41]. An ABTS tablet was dissolved in water and reacted with a 2.45 mM potassium persulfate solution to obtain a 7 mM solution. The mixture was incubated for 12 h at 4 °C to generate free radicals, and then a 10 μL sample was reacted with the ABTS solution (190 μL) at room temperature for 10 min in the dark. After incubation, the absorbance was recorded at 734 nm with a UV/Vis spectrophotometer. ABTS radical scavenging activity (%) was presented according to the following formula:$$ABTS\;radical\;scavenging\;activity\left(\%\right)=\frac{{Abs}_{control}-{Abs}_{sample}}{{Abs}_{control}}\times 100$$
where Abs_sample_ is the absorbance of the tested sample and Abs_control_ is the absorbance of the control after 10 min reaction.

### 2.8. Analysis of VOCs Using Gas Chromatography Time-of-Flight Mass Spectrometry (GC-TOF-MS)

Solid-phase microextraction (SPME) analysis was performed following an established protocol [42] using GC coupled with TOF-MS (LECO Pegasus GC HRT, Leco Corporation, St. Joseph, MI, USA). A freeze-dried sample (10 mg) was mixed with 2 mL of 30% NaCl as the saturated solution, and 2 μL of 0.2 mg/mL 3-pentanol was used as the internal standard. The SPME holder equipped with a 50/30 μm fiber (DVB/CAR/PDMS, model 57348-U, Supelco, Bellefonte, PA, USA) was used for sampling. Before use, the fiber was conditioned at 250 °C for 5 min. Aromatic compounds were extracted through SPME fiber at 70 °C and a stirring rate of 500 rpm for 20 min. After stirring, GC analysis was performed using a capillary column (Rtx-5MS column, 30 m length, 0.25 mm diameter, 0.25 μm thickness, 5% diphenyl, 95% dimethyl polysiloxane, Restek, Bellefonte, PA, USA) under split mode (30:1). The front inlet temperature was set at 240 °C and the transfer line temperature was maintained at 250 °C. The mass range was from 36 to 450. The ion source temperature was kept at 250 °C with an ionization voltage of 70 eV and the helium was used as carrier gas at a flow rate of 1 mL/min. The oven temperature program was 60–150 °C at the rate of 13 °C/min and then 150–180 °C at the rate of 8 °C/min, and it was then programmed to 180–200 °C at the rate of 10 °C/min and finally reached 245 °C at the rate of 30 °C/min. It was then held there for 3 min.

### 2.9. Statistical Analysis

All data are expressed as the mean ± standard error (SE) of three biological replicates (*n* = 3). The data were checked for normality and subjected to a one-way analysis of variance (ANOVA) with SPSS Statistics (version 26.0, SPSS Inc., Chicago, IL, USA) based on Duncan’s test (*p* < 0.05, 95% confidence interval). The statistical results were represented by asterisks (*, **, and ***), which correspond to *p* values of <0.05, <0.01, and <0.005, respectively. The heatmap for Pearson’s correlation analysis was generated using Metabo Analyst 5.0.

## 3. Results and Discussion

### 3.1. Selected Plant Growth Attributes

During the cultivation period, environmental conditions (temperature, relative humidity, and CO_2_ concentration) were well maintained (Table 1). Significant differences were observed between cultivation periods for plant morphological properties, as shown in Figure 1, as well as plant heights and weights, as detailed in Table 2. The weights of AR and OBP were remarkably varied following the cultivation period. Flowering started from 70 DAS in AR (Figure 1) and 90 DAS in OBP (Appendix A), and these influenced their growth rate calculated based on the weight change. At a particular time (pre-flowering), the growth rate dramatically increased from 0.39 (30–60 DAS) to 2.12 (60–70 DAS) in AR and from 0.58 (60–70 DAS) to 0.85 (70–80 DAS) in OBP. Meanwhile, the other plants (MO and SO) that did not flower in this experiment showed a consistent increase in growth rate of 0.01 (0–30 DAS) to 1.94 (70–80 DAS) in MO and 0.05 (0–30 DAS) to 0.81 (70–80 DAS) in SO, respectively. These observations were consistent with the previous literature that explained that the plant’s growth rate can be calculated by weight change. After the planting, small plant sizes start to increase dramatically and stabilize after the flower induction. Therefore, the transition from the vegetative phase to the reproductive phase can be recognized by the growth curve by finding the initial of the vegetative phase (flowering stage) [43,44].

Based on their phenological properties, four classes (S1, seedling stage; S2, vegetative stage; S3, pre-flowering stage; S4, flowering stage) were divided, and for MO and SO, which did not appear to transition from the vegetative stage to reproductive stage, sub-classes for vegetative stages were used.

### 3.2. Estimation of Total Phenolics Content, Rosmarinic Acid Content, and Their Influence on Antioxidant Activity

Based on the HPLC-DAD analysis, there was one major compound detected commonly at 280 nm across all plant species. The main peak was identified as the RA by its specific MS and MS^2^ spectrum derived from its fragmentation pattern (Figure 2) through UPLC-Tandem MS analysis. Based on the MS and MS^2^ spectrum, the fragmentation pattern (Figure 2C) showed a peak at *m*/*z* 359 as the precursor ion and MS^2^ spectrum exhibited the ions at *m*/*z* 197, 179, 161, and 135, corresponding to [caffeic acid (C_9_H_8_O_4_) − H − CO_2_]^−^, [caffeic acid (C_9_H_8_O_4_) − H − H_2_O]^−^, [caffeic acid (C_9_H_8_O_4_) − H]^−^, and [quinic acid (C_7_H_12_O_6_) − H]^−^, respectively.

Through qualitative analysis of selected *Lamiaceae* plants, RA was observed as the main compound, as described in Figure 3. Previously, phenolics including RA have had a high association with antioxidant activity (AOA), and this can result in the presence of diverse pharmacological properties in plants [45]. Therefore, spectroscopic measurement was conducted for TPC and HPLC analysis detected at 280 nm was performed for quantification of RA (Figure 4). Additionally, DPPH radical scavenging activity (DPPH) and ABTS radical scavenging activity (ABTS) were evaluated to assess their influence on antioxidant activities (Figure 5). The distribution of TPC and RAC in *Lamiaceae* plants varied significantly depending on plant species and their growth progression, as shown in Figure 4.

In the case of AR, RAC increased during the vegetative phases (S1, 18.3 ± 4.6 mg/g; S2, 35.0 ± 6.8 mg/g; S3, 71.4 ± 5.2 mg/g) and decreased during the flowering phase (S4, 65.1 ± 10.1 mg/g). However, there were no significant differences in TPC observed from 60 DAS (Figure 4A,B). The antioxidant activity, which was approximately 12.4 ± 2.2% for DPPH and 17.2 ± 1.8% for ABTS at 30 DAS, steadily increased and showed the highest DPPH (28.8 ± 2.3%) and ABTS (30.5 ± 1.4%) at 80 DAS at a sample concentration of 100 µg/mL and 50 µg/mL, as shown in Figure 5A. This low trend concordance was once again confirmed through correlation analysis (Figure 7A). A strong correlation coefficient (*r*) of 0.7181 (*p* < 0.01) was observed between TPC and RAC, and both were positively affecting AOA. In the case of RAC, the *r* values between DPPH (*r* = 0.8291, *p* < 0.005) and ABTS (0.8148, *p* < 0.005) were similar, while TPC had a more significant positive correlation with ABTS (*r* = 0.9564, *p* < 0.005) than with DPPH (*r* = 0.7181, *p* < 0.005). A similar pattern showing a different tendency between TPC and RAC was also observed from SO. There was no dramatic change in TPC. However, in the case of RAC, it was found to increase significantly (*p* < 0.05) approximately 3.88 times during the transition from S1 (30 DAS) to S2-1 (60 DAS), and this content was maintained during the vegetative phases (S2, 70–80 DAS). Not only TPC results but these secondary metabolites are also considered to affect the AOA of SO. In terms of AOA, DPPH activity was highly determined at 70 and 80 DAS with activity values of 62.9 ± 0.9 and 58.8 ± 1.5% at a sample concentration of 100 µg/mL, as depicted in Figure 5D. Meanwhile, a non-significant difference (*p* < 0.05) was observed in ABTS. A higher correlation coefficient between RAC and DPPH (*r* = 0.8840, *p* < 0.005) was observed than between TPC and DPPH (*r* = 0.6420, *p* < 0.05) (Figure 7D). Therefore, it was explained that RAC is a more important factor than other phenolics when determining the AOA of SO. In the case of MO, both TPC and RAC showed a moderate–strong correlation coefficient with growth parameters (weight and height) during the growth progression, indicating a consistent increase (Figure 4). TPC and RAC increased from 232.5 mg GAE/g and 29.4 mg/g in S1 (30 DAS) to 323.7 mg GAE/g and 120.3 mg/g in S2-3 (80 DAS), following a similar trend that led to a moderate correlation between TPC and RAC (*r* = 0.6292, *p* < 0.05) (Figure 6B). Consequently, the positive influence of both TPC and RAC on AOA was depicted. TPC showed a higher correlation with DPPH (*r* = 0.9011, *p* < 0.005) and ABTS (*r* = 0.9220, *p* < 0.005) than RAC (DPPH: *r* = 0.7843, *p* < 0.005 and ABTS: *r* = 0.7423, *p* < 0.01) (Figure 7B). Therefore, it has been proven that the contribution of TPC to AOA is higher than that of RA.

Similar results indicating a higher association of TPC to AOA were observed for OBP. All factors were found to have a significantly strong correlation, exhibiting sufficient evidence (*p* < 0.05). During the cultivation period, TPC and RAC increased from 153.7 mg GAE/g and 30.2 mg/g in S1 (30 DAS) to 369.0 mg GAE/g and 224.2 mg/g in S3 (80 DAS), and these compound changes strongly influenced AOA (Figure 5C). DPPH and ABTS measured at 100 μg/mL and 50 μg/mL increased from 16.3 ± 0.0% and 39.4 ± 4.5% in S1 (30 DAS) to 61.1 ± 1.0% and 65.0 ± 0.7% in S3 (80 DAS).

Natural antioxidants can delay or inhibit lipid oxidation by suppressing the initiation or propagation of oxidative chain reactions [46]. Phenolic compounds are widely recognized for their desirable antioxidant properties. They act as reducing agents, single oxygen quenchers, hydrogen donors, and chelating agents of metal ions [47,48]. The antioxidant activity of phenolics typically depends on the arrangement and substitution pattern of hydroxyl groups, making the presence and proportion of active compounds crucial [46]. Although all plants in this study contained RA as the main component, their contributions to antioxidant activity were differently confirmed. According to the influence on the AOA of TPC and RAC, expressed as correlation coefficients, they could be divided into two major groups. The dominant group of the combination of phenolics included AR and SO, while the dominant group of RAC alone included OBP and MO. This was explained by the compound profiles and their respective antioxidant capacity in each plant species.

For secondary metabolites of AR, numerous phenolics such as 4-hydroxybenzoic acid, chlorogenic acid, caffeic acid, cinnamic acid, and flavonoids including quercetin, rutin, kaempferol, tilianin, and acacetin have been reported as well as RA [49,50]. In particular, the amount and proportion of major phytochemicals such as tilianin, acacetin derivatives, and RA present in AR were found to vary [51,52]. It was explained by the increase in RA and decrease in other components (quercetin, tilianin, acacetin, etc.) during the flower development process [49,52]. The higher phenolic levels in flowers produced stronger antioxidant activity than in the stems and leaves, resulting in an improvement of the biological activities of the reproductive stage [53]. Therefore, the highest antioxidant activity observed at 70 DAS and the significant contribution of RA to this activity can be well explained by the entry into the reproductive stage. In the case of SO, the previous literature reported that carnosol [54], quercetin derivative [55], and campherol [56] are also included in SO as secondary metabolites. Furthermore, abietane-type diterpenoids, such as carnosic acid and carnosol, along with RA, significantly contribute to the antioxidant activities of SO [57]. However, during the transition from the vegetative stage to the reproductive stage, there is an opposite trend in the accumulation of active compounds, with an increase in phenolic diterpenes (such as carnosic acid and carnosol) and flavonoids (such as apigenin, hispidulin, cirsimaritin, and naringin) and a decrease in RA [58]. Additionally, Lu and Yeap Foo [59] explained that flavonoids exhibit comparatively weaker antioxidant activity (AOA), while RA derivatives display stronger AOA, than Trolox in SO. During the experiment, SO did not enter the flowering period, so it was considered as a vegetative state with RA being dominant. Therefore, it was explained that the effect of RA on antioxidant activity was greater.

Meanwhile, MO and OBP showed other antioxidant active compounds’ effects on AOA. MO is known for containing RA as its main compound, along with other phenolic compounds including ferulic acid, gallic acid, chlorogenic acid, syringic acid, p-coumaric acid, and caffeic acid, which are present in high proportions [60]. Caffeic acid, in particular, is found as another prominent phenolic in MO [61,62] and is more effective in inhibiting lipid oxidation and oil-in-water emulsion oxidation activities than RA [63]. Phenolic compounds as well as flavonoids are important active compounds of MO, although their profiles vary depending on the variety. Abdellatif et al. [64] explained that several flavonoids, including quercetin, luteolin, and kaempferol, are dominant active compounds, with relative contents exceeding 1%; among these, quercetin has excellent DPPH scavenging ability comparable to RA. Therefore, the relatively high amount of flavonoids in MO, showing AOA, was the reason for the lower correlation between RAC and AOA than between TPC and AOA [65]. This similar trend is also observed in OBP, which is reported to contain other phenolics such as caffeic acid, caftaric acid, chicoric acid, etc. [66]. Their distribution is differently reported following the varieties [67] but, in the case of OBP used in this experiment, RA dominated the composition, with no other major components detected at a wavelength of 280 nm, which is generally used for detecting phenolics (Figure 3C). While the color of OBP is typically attributed to anthocyanins, their content and composition could be responsible for the red and blue pigmentation in plants [66]. However, there have been reports that explain the low contribution of anthocyanins to the AOA of basil. Therefore, focusing on the presence of a high amount of RA was sufficient to explain the high correlation between RAC-AOA and TPC-AOA [66,68,69].

### 3.3. Profiling of VOCs and Their Distribution

*Lamiaceae* plants have a small number of primary ingredients that contribute significantly to their characteristic fragrance by comprising more than 40% of the VOCs’ content [70]. These VOCs have gained recognition as promising metabolites, known for their safety and applications in antioxidant, antibacterial, and antimutagenic activities with non-toxicity. Ensuring the uniformity of VOCs is essential for maintaining quality due to their high variability. For the VOCs, not only the genetic variability, but also plant parts, different developmental stages, and environmental conditions regulate the VOCs content [21]. The profile of VOCs is strongly influenced by plant species such as thyme (*Thymus vulgaris*) of thymol [71], rosemary (*Salvia rosmarinus*) of α-pinene [72], and peppermint (*Mentha piperita*) of carvone [73,74]. The essential oil content was influenced by the growth stage in thyme [26,27], oregano (*Origanum vulgare*) [28], and rosemary [23,24], and these changes have also been confirmed to influence the antimicrobial [30] and AOA [31] of the *Lamiaceae* plants. To profile the VOCs, an SPME analysis was conducted, and the results varied among the different plant species (AR: Table 3; MO: Table 4; OBP: Table 5; SO: Table 6). The relative abundance of identified peaks detected by GC-MS was calculated as a percentage of the total area of identified peaks, with peak areas normalized using the internal standard, 3-pentanol. Significant fluctuations in the VOCs of the selected plants based on sampling dates were observed. 

One commonality among all plant species was the presence of (*E*)-2-hexanal, a naturally occurring volatile C6 aliphatic aldehyde compound [75]. It acts as a germination inhibitor [76] and accumulates as a product of lipid peroxidation during the germination process [77]. These aldehydes result from the degradation of hydroperoxides, which are produced from the conversion of unsaturated fatty acids compositing the main membrane of lipids susceptible to peroxidation into free radicals and C6/C9 hydroperoxides [78]. The accumulation of (*E*)-2-hexanal at 30 DAS indicated the germination process, while its decrease after 60 DAS signified the transition to the growth stages. Additionally, for the dominant volatile compounds, all the plant species showed an increased tendency in the initial vegetative stages, followed by a decrease in total VOCs content over time, coinciding with a decrease in the main component. 

In the case of AR (Table 3), a total of 16 compounds were identified and these consist of aldehydes (2), fatty alcohol (1), monoterpenes (4), sesquiterpenes (3), terpene alcohols (2), and phenylpropenes (4). A significant effect of the growth stage on total VOCs was observed by varied total VOCs content. It varied from 71.7 ± 7.0 ng/mg at 70 DAS to 243.4 ± 21.3 ng/mg at 60 DAS. As depicted in Appendix A, the phenylpropene group constituted a substantial proportion, ranging from 96.7% to 97.8% of the total VOCs. The dominance was primarily attributed to the presence of estragole, the most abundant compound. Changes in estragole content influenced the overall total content, with a notable increase up to 60 DAS (S2), followed by a sharp decrease. AR proved to be a rich source of estragole, as indicated in Table 3. Estragole is a well-known aroma ingredient widely used in food products as a flavoring agent [79], and it accounted for a significant proportion, ranging from 46.7% to 94.6% in AR [80,81].

In statistical analysis, estragole exhibited a very strong positive correlation with the total VOCs content, with a correlation coefficient (*r*) of 0.9999. The data presented indicated that the total VOCs content (S2: 242.4 ± 21.3 ng/mg; S1: 207.3 ± 14.3 ng/mg; S4: 94.2 ± 11.7 ng/mg; S3: 71.7 ± 7.0 ng/mg) tended to decrease over time in tandem with the diminishing estragole content (Figure 6A). As a result, the maximized stage (S2) was considered the most efficient for obtaining estragole. EOMT (eugenol-O-methyltransferase) activity and eugenol accumulation have a significant correlation and the decrease in the transcript expression level of EOMT with leaf age [82] was expected to be responsible for the decrease in the total amount of VOCs.

In MO (Table 4), 18 VOCs were identified, including aldehydes (3), monoterpenes (4), sesquiterpene (1), terpene alcohols (3), phenylpropene (1), ketones (2), furan (1), ester (1), lactone (1), and fatty alcohol (1). Monoterpenes composed of isoneral, (*Z*)-neral (citral B), geranial (citral A), and methyl geraniate accounted for approximately 87% of the VOCs (Figure 6B) and geranial was identified as the major component of MO (Table 4). Specifically, the primary composition of VOCs in S2 stage MO, which exhibited the highest VOCs content, consisted of geranial (66.5%) and (*Z*)-neral (30.1%). The combined total of these two main components represented a significant percentage in all MO samples, regardless of the stages, ranging between 87.0% and 96.8% (Figure 6B). Consequently, the variation in total VOCs content across the stages was primarily attributed to changes in geranial and (*Z*)-neral content (as shown in Table 4), and the order from highest to lowest content was as follows: S2-1 (60 DAS; 215.6 ± 34.1 ng/mg) > S1 (30 DAS; 166.0 ± 4.2 ng/mg) > S2-2 (70 DAS; 35.2 ± 12.6 ng/mg) > S2-3 (80 DAS; 20.6 ± 11.1 ng/mg). In the case of MO, some previous reports have referred to the major components of VOCs of MO as the geranial, (*Z*)-neral, citronellal, (*E*)-caryophyllene, caryophyllene oxide, geraniol, etc. [83]. Among these diverse monoterpenes, monoterpene aldehydes (geranial and neral) were identified as the main component of MO. Although MO exhibited relatively higher levels of geranial compared to (*Z*)-neral, both compounds made significant contributions. These two stereoisomers of citral (mixture of geranial and neral) are also found in citrus fruits, lemongrass, and gingers to determine quality [84], and were highly quantified in the S2-1 stage and then declined. Total VOCs content was statistically concerned with geranial and (Z)-neral with correlation coefficients of 0.9866 (*p* < 0.005) and 0.9980 (*p* < 0.005), respectively. Consequently, to obtain MO with a rich VOC content, the optimal stages were S1 (30 DAS) and S2-1 (60 DAS).

In contrast to the two previously discussed plant species, OBP and SO did not exhibit remarkable major VOCs. In OBP, 32 VOCs, which included aldehydes (4), monoterpenes (9), sesquiterpenes (6), terpene alcohols (8), phenylpropenes (4), and fatty alcohol (1), were found, as described in Table 5. Their distribution varied across the different growth stages, with S2-1 (60 DAS; 259.0 ± 14.9 ng/mg) containing the highest VOCs, followed by S1 (30 DAS; 221.6 ± 17.2 ng/mg), S3 (80 DAS; 48.5 ± 7.3 ng/mg), and S2-2 (70 DAS; 11.1 ± 3.8 ng/mg), respectively. Notably, the terpene alcohols and phenylpropenes displayed opposite patterns as the days progressed. The proportion of phenylpropenes steadily decreased from 93.5% to 47.9%, while the terpene alcohols’ portion consistently increased from 3.1% to 37.1% over time (Appendix A). The primary aromatic compounds in OPB were eugenol, methyl eugenol, and linalool (Table 5). During the S1 (30 DAS) and S2-1 (60 DAS) stages, methyl eugenol was the major component, accounting for 58.2–74.9%, with lower amounts of eugenol (17.9%) and linalool (1.9%). However, during the S2-2 (70 DAS) stage, eugenol, and linalool content increased, reaching 35.6% and 31.9%, respectively, making them the most abundant compounds (see Figure 6C). This composition of OPB consisted of reported paper [85]. Methyl eugenol was the dominant constituent of OBP, followed by eugenol. 

The strong correlation between these two phenylpropenes (*r* = 0.8458, *p* < 0.005) suggests the possibility of conversion from eugenol to methyl eugenol through the catalysis of the EOMT gene [86]. Both eugenol and methyl eugenol exhibited a very strong correlation with total VOCs content (eugenol: *r* = 0.9451, *p* < 0.005; methyl eugenol: *r* = 0.9678, *p* < 0.005, respectively). Meanwhile, the other major compound, linalool, which is one of the terpene alcohols, showed a weak correlation with total VOCs (*r* = 0.2844, *p* < 0.005). Therefore, eugenol and methyl eugenol were revealed as the principal factors to determine total VOCs emission in OBP. The highest content of phenylpropene compounds was observed during the initial vegetative stage (S2-1). This finding is consistent with the results of Renu et al. [87], who reported an increase in total VOC content during the juvenile stage (S2-1), followed by a rapid reduction during the pre-flowering and flowering stages in Ocimum species.

In SO, 34 VOCs, including aldehydes (3), monoterpenes (17), sesquiterpenes (2), terpene alcohols (9), phenylpropenes (2), and fatty alcohol (1), were found, as described in Table 6. Several compound groups constituted a large portion (more than 10%) of the total VOCs. Monoterpenes (63.4–79.2%), sesquiterpenes (10.1–15.6%), and terpene alcohols (9.1–18.2%) were associated with the aroma of SO (Table 6 and Appendix A). Unlike AR and MO, where several components determine the total amount, the sum of some of the main compounds, such as (*Z*)-thujone (28.4–43.1%), camphor (15.5–29.4%), and humulene (8.8–13.7%), accounted for approximately only half (Figure 6D). The total VOCs content varied across different growth stages, including S2-1 (60 DAS; 128.5 ± 8.5 ng/mg), S1 (30 DAS; 94.5 ± 7.5 ng/mg), S2-3 (80 DAS; 75.9 ± 12.8 ng/mg), and S2-2 (70 DAS; 61.5 ± 3.4 ng/mg) (Table 6). Interestingly, there were no remarkable major VOCs in SO. Without overwhelming compounds, diverse volatile compounds were evenly distributed, and relatively predominant volatile compounds (making up more than 8% of the total VOCs content) included two monoterpenes, (*Z*)-thujone and (−)-camphor, and one sesquiterpene, humulene. Among three high-ratio components, in correlation analysis, only (−)-camphor showed a very strong correlation coefficient (*r* = 0.9638, *p* < 0.005), while others ((*Z*)-thujone and humulene) exhibited comparatively lower correlation coefficients (*r* = 0.7560 and 0.7322, respectively). The aromatic properties of SO have been reported to be composed of thujone diastereomeric forms (α-thujone and β-thujone), 1,8-cineol, camphene, humulene, α-pinene, limonene, and borneol, among others, varying with the species. These compounds make SO a commonly used savory food flavoring in the form of dried leaves and essential oil [88,89]. Therefore, the worth of the S1 and S2-1 stages was evaluated to be higher due to higher total VOCs content.

For the *Lamiaceae* plants, VOCs are important quality indicators. However relatively little is known about the effect of growth stages on VOCs distribution. Some studies have provided evidence that some VOCs stimulate seed germination and seedling stages [90]. Thus, young stages show dramatic changes in VOCs emissions, and leaf ontogeny can greatly influence VOCs production [91]. This study indicated that variations of the VOCs emitted by *Lamiaceae* species significantly depend on the growth stage. Although the main components and their proportions were all different, the total VOCs of four *Lamiaceae* plants showed the highest content while passing from the S1 to S2 stage. Consequently, it is suggested that the seedling stages of *Lamiaceae* plants under specific periods (leaf development to stem elongation) can serve as rich sources of VOCs. 

**Figure 7 antioxidants-12-01909-f007:**
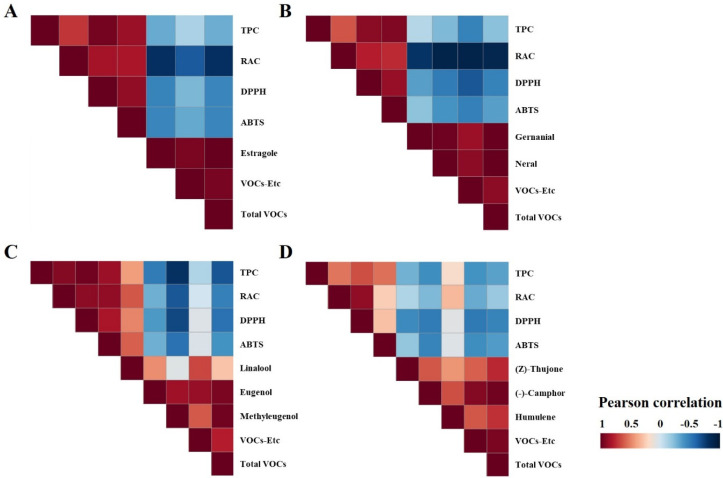
Heatmap illustrating the correlation analysis of (**A**) AR, (**B**) MO, (**C**) OBP, and (**D**) SO generated by Pearson’s correlation coefficient (*r*) between phenolic relative factors, antioxidant activities, and VOCs. TPC, total phenolics content; RAC, rosmarinic acid content; DPPH, DPPH radical scavenging activity at 100 µg/mL; ABTS, ABTS radical scavenging activity at 50 µg/mL; VOCs-Etc, VOCs content excluding main components.

## 4. Conclusions

Overall, it could be concluded that harvest timing affects growth pattern, phytochemicals content, and even antioxidant activities of *Lamiaceae* species including *Agastache rugosa* (AR), *Melissa officinalis* (MO), *Ocimum basilicum Purpurascens* (OBP), and *Salvia officinalis* (SO). Due to the high proportion of rosmarinic acid (RA) in the plants of Lamiaceae, it has been revealed as a significant factor contributing to antioxidant activities through statistical analysis among phenolic compounds. Furthermore, the pattern of RA content varied depending on whether they had flowered or not. In the case of flowering species (AR and OBP), RAC increased dramatically when transitioning from the vegetative stage to the reproductive stage. Meanwhile, the non-flowering species did not show a rapid increase in TPC and RAC. In terms of VOCs, significant differences were observed in components following the species. For AR and MO, there was a specific compound group that accounted for the largest portion, phenylpropene (estragole) and monoterpenes (citrals and methyl geraniate). In OBP and SO, various component groups constituted the emitted VOCs. Terpenoids including (Z)-thujone, camphor, and humulene distributed to the flavor of SO. Changes in the combination of phenylpropenes (eugenol and methyl eugenol) and terpene alcohol (linalool) determined the flavor of OBP. Although the component composition and ratio were different, the highest content fragrance components were common in all plants at the time of transition from S1 to S2. In conclusion, it is recommended to utilize the pre-flowering stage to obtain a high content of RA and post-germination plants for VOCs.

## Figures and Tables

**Figure 1 antioxidants-12-01909-f001:**
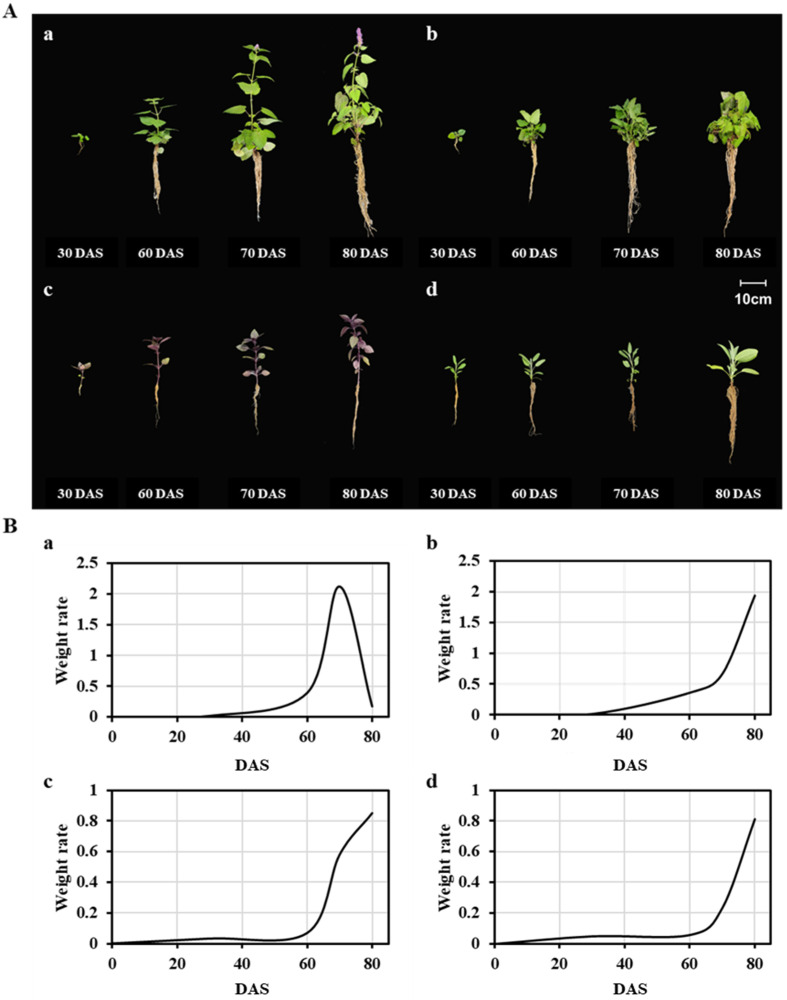
(**A**) Morphological characters and (**B**) relative growth rate calculated based on the weight of plants (**a**: AR; **b**: MO; **c**: OBP; **d**: SO) collected on 30, 60, 70, and 80 DAS.

**Figure 2 antioxidants-12-01909-f002:**
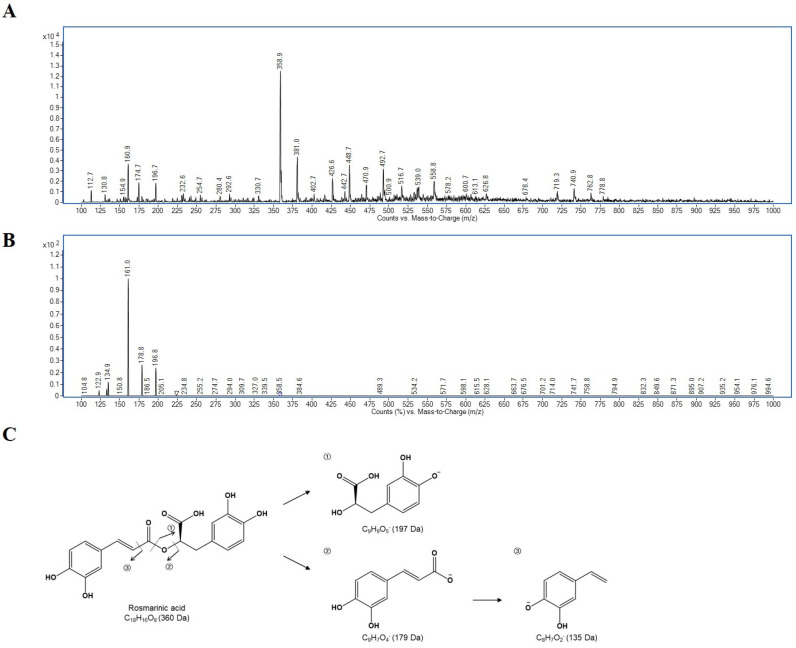
Chromatogram of RA at m/z 359 under (**A**) the negative ion mode full MS spectra and (**B**) MS2 spectra and (**C**) fragmentation pattern of RA.

**Figure 3 antioxidants-12-01909-f003:**
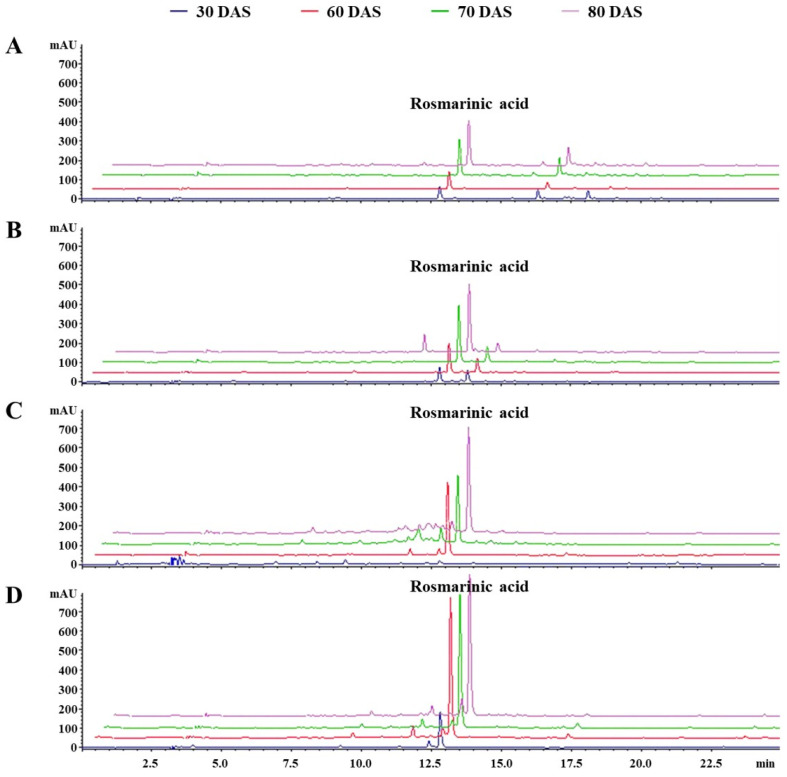
HPLC chromatogram of (**A**) AR, (**B**) MO, (**C**) OBP, and (**D**) SO observed at 280 nm. Different line colors of the chromatogram indicated harvest time according to the DAS.

**Figure 4 antioxidants-12-01909-f004:**
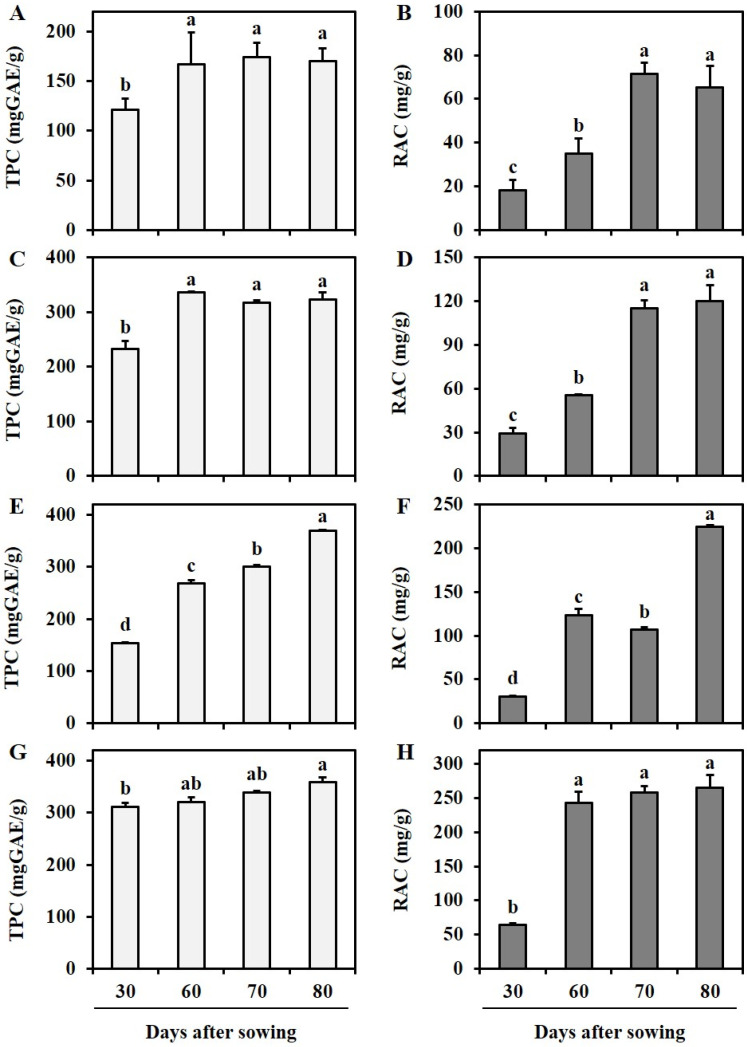
Changes in TPC (mg GAE/g) and RAC (mg/g) from AR (**A**,**B**), MO (**C**,**D**), OBP (**E**,**F**), and SO (**G**,**H**) at elevated DAS. Values were expressed as mean ± standard error (SE). Different lower letters (a–d) indicate statistically significant differences of TPC and RAC among DAS following Duncan’s multiple-range test method (*p* < 0.05).

**Figure 5 antioxidants-12-01909-f005:**
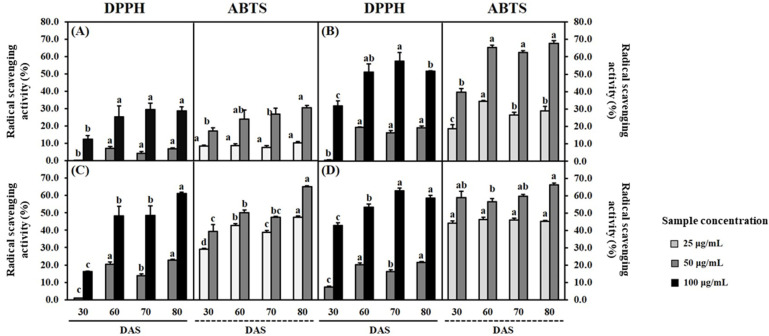
DPPH radical scavenging activity (50 and 100 μg/mL) and ABTS radical scavenging activity (25 and 50 μg/mL) of (**A**) AR, (**B**) MO, (**C**) OBP, and (**D**) SO at elevated DAS. Values were expressed as mean ± standard error (SE). Statistical differences of antioxidant activities among DAS were shown as lowercase letters (a–c) using Duncan’s multiple-range test method (*p* < 0.05).

**Figure 6 antioxidants-12-01909-f006:**
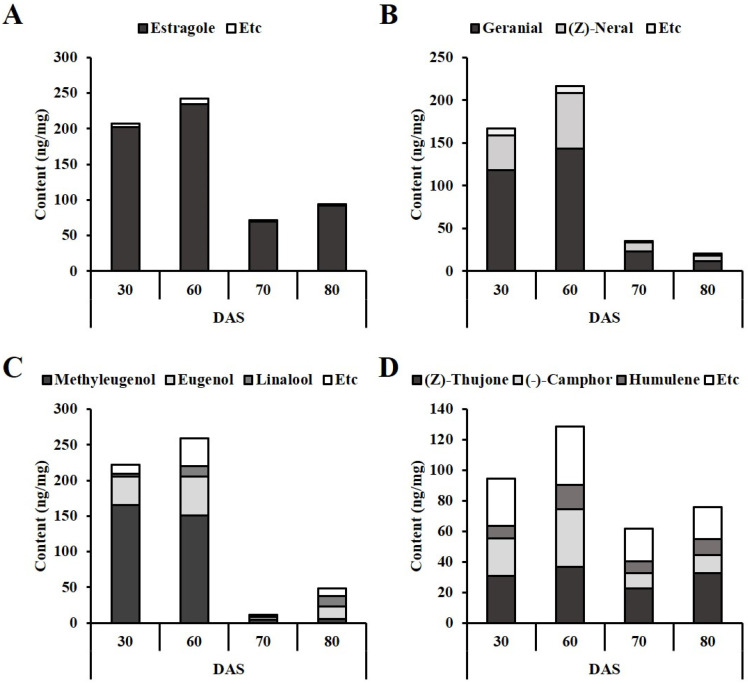
Effect of cultivation period on main VOCs of (**A**) AR, (**B**) MO, (**C**) OBP, and (**D**) SO following the elevated DAS. The main components (those with a high proportion in terms of total quantity) and Etc (representing the content of the minor components after subtracting main compounds content from the total quantity) were indicated.

**Table 1 antioxidants-12-01909-t001:** Climate data of the greenhouse during the experiment controlled with actuator equipment and monitoring system.

Cultivation Period	Date	Temperature			RH ^1^ (%)	CO_2_ Conc. (ppm)
		Mean. (°C)	Max. (°C)	Min. (°C)		
30–40 DAS	12–21 February	18.7	30.2	14.0	50.6	439.8
40–50 DAS	22 February–31 March	18.7	30.1	15.3	53.1	430.1
50–60 DAS	4–13 March	19.6	31.1	15.1	51.0	425.2
60–70 DAS	14–23 March	20.5	32.0	12.6	50.6	414.1
70–80 DAS	24 March–2 April	19.9	29.5	14.5	55.0	428.7

^1^ relative humidity.

**Table 2 antioxidants-12-01909-t002:** Plant height and weight change of AR, MO, OBP, and SO. Data were presented as mean ± standard error.

Species	Harvest Time	Description	Stage	Plant Weight (g)	Plant Height (cm)
AR ^1^	30 DAS ^5^	Seedling stage	S1	0.56 ± 0.01 d	7.67 ± 0.33 d
	60 DAS	Vegetative stage	S2	12.33 ± 0.42 c	63.33 ± 1.76 c
	70 DAS	Pre-flowering stage	S3	33.57 ± 0.41 b	80.00 ± 1.15 b
	80 DAS	Flowering stage	S4	35.30 ± 0.10 a	88.33 ± 0.88 a
MO ^2^	30 DAS	Seedling	S1	0.41 ± 0.02 d	11.33 ± 0.88 d
	60 DAS	Vegetative stage	S2-1	11.24 ± 0.20 c	40.33 ± 1.45 c
	70 DAS	Vegetative stage	S2-2	17.91 ± 1.27 b	61.33 ± 0.88 b
	80 DAS	Vegetative stage	S2-3	37.33 ± 1.19 a	66.00 ± 1.15 a
OBP ^3^	30 DAS	Seedling	S1	0.95 ± 0.01 d	13.50 ± 0.29 d
	60 DAS	Vegetative stage	S2-1	3.00 ± 0.17 c	34.33 ± 1.20 c
	70 DAS	Vegetative stage	S2-2	8.81 ± 0.18 b	50.00 ± 1.15 b
	80 DAS	Pre-flowering stage	S3	17.33 ± 0.33 a	55.33 ± 1.45 a
SO ^4^	30 DAS	Seedling	S1	1.49 ± 0.04 d	27.00 ± 0.58 c
	60 DAS	Vegetative stage	S2-1	3.23 ± 0.03 c	29.67 ± 1.33 c
	70 DAS	Vegetative stage	S2-2	5.57 ± 0.07 b	34.67 ± 1.45 b
	80 DAS	Vegetative stage	S2-3	13.67 ± 0.67 a	47.00 ± 1.73 a

Different lowercase letters (a–d) indicate significant differences of weight and height among DAS following Duncan’s multiple-range test (*p* < 0.05). ^1^
*Agastache rugosa*, ^2^
*Melissa officinalis*, ^3^ *Ocimum basilicum* Purpurascens, ^4^ *Salvia officinalis*, ^5^ days after sowing.

**Table 3 antioxidants-12-01909-t003:** VOCs’ profiling change (ng/mg) of AR following the DAS.

No.	Compound	RI ^1^	Classification	DAS ^2^			
				30	60	70	80
1	(*E*)-2-Hexanal	857	Aldehyde	1.52 ± 0.26 a	0.16 ± 0.02 b	0.23 ± 0.06 b	0.19 ± 0.04 b
2	1-Octen-3-ol	981	Fatty alcohol	0.13 ± 0.03 b	0.27 ± 0.02 ab	0.23 ± 0.01 ab	0.32 ± 0.05 a
3	Limonene	1031	Monoterpene	0.50 ± 0.07 ab	1.10 ± 0.39 a	0.16 ± 0.01 b	0.23 ± 0.03 ab
4	4-Methyl benzaldehyde	1079	Aldehydes	0.51 ± 0.04 a	0.44 ± 0.11 a	0.02 ± 0.00 b	0.03 ± 0.01 b
5	Linalool	1101	Terpene alcohol	ND ^2^	0.10 ± 0.00 ab	0.15 ± 0.04 a	0.02 ± 0.00 b
6	Isopulegone	1179	Monoterpene	0.04 ± 0.00 a	0.04 ± 0.01 a	0.02 ± 0.00 a	0.03 ± 0.00 a
7	Estragole	1198	Phenylpropene	202.55 ± 7.88 a	234.33 ± 11.97 a	69.99 ± 4.21 b	92.05 ± 6.40 b
8	(*Z*)-Neral	1243	Monoterpene	ND	0.84 ± 0.11 a	0.16 ± 0.04 b	0.16 ± 0.01 b
9	p-Chavicol	1257	Phenylpropene	0.04 ± 0.00 a	0.02 ± 0.00 b	ND	ND
10	Geranial	1271	Monoterpene	ND	1.46 ± 0.19 a	0.29 ± 0.09 b	0.26 ± 0.02 b
11	Anethole	1289	Phenylpropene	1.14 ± 0.31 b	2.20 ± 0.04 a	0.20 ± 0.02 c	0.31 ± 0.04 c
12	Methyleugenol	1408	Phenylpropene	0.04 ± 0.01 b	0.04 ± 0.00 b	0.09 ± 0.01 a	0.06 ± 0.01 ab
13	Caryophyllene	1423	Sesquiterpene	0.42 ± 0.05 ab	1.07 ± 0.20 a	0.10 ± 0.03 b	0.48 ± 0.27 ab
14	(*E*)-β-Famesene	1459	Sesquiterpene	0.07 ± 0.04	ND	ND	ND
15	δ-Cadinene	1527	Sesquiterpene	0.12 ± 0.01 ab	0.23 ± 0.04 a	0.05 ± 0.01 b	0.10 ± 0.04 b
16	Spathulenol	1579	Terpene alcohol	0.07 ± 0.00 a	0.07 ± 0.01 a	ND	ND
		Total		207.13 ± 14.31 a	242.37 ± 21.31 a	71.71 ± 6.97 b	94.24 ± 11.65 b

Values were expressed as mean ± standard error and experiments were conducted with repetitive analyses. An ANOVA test was performed with Duncan’s multiple-range test using the SPSS program and the result was expressed as different letters (a–c). ^1^ RI: retention index; ^2^ ND: not detected.

**Table 4 antioxidants-12-01909-t004:** VOCs’ profiling change (ng/mg) of MO following the DAS.

No.	Compound	RI ^1^	Classification	DAS			
				30	60	70	80
1	(*E*)-2-Hexanal	857	Aldehyde	0.31 ± 0.07 a	0.25 ± 0.02 ab	0.14 ± 0.02 ab	0.08 ± 0.00 b
2	1-Octen-3-ol	981	Fatty alcohol	0.03 ± 0.01 b	0.04 ± 0.00 b	0.07 ± 0.02 b	0.13 ± 0.01 a
3	6-Methyl-5-hepten-2-one	986	Ketones	0.26 ± 0.00 ab	0.47 ± 0.09 a	0.19 ± 0.04 b	0.13 ± 0.02 b
4	(*E,E*)-2,4-Heptadienal	1016	Aldehyde	0.04 ± 0.01 c	0.02 ± 0.00 c	0.16 ± 0.02 b	0.27 ± 0.03 a
5	4-Methyl benzaldehyde	1079	Aldehyde	1.33 ± 0.33 a	0.42 ± 0.01 b	0.04 ± 0.00 b	0.05 ± 0.00 b
6	Linalool	1101	Terpene alcohol	0.25 ± 0.03 a	0.11 ± 0.01 b	0.03 ± 0.00 c	0.02 ± 0.01 c
7	Isoneral	1165	Monoterpene	0.37 ± 0.01 a	0.49 ± 0.10 a	0.11 ± 0.01 b	0.14 ± 0.01 b
8	Rose furan oxide	1178	Furans	0.37 ± 0.01 a	0.49 ± 0.10 a	0.02 ± 0.01 b	0.01 ± 0.00 b
9	Estragole	1198	Phenylpropene	2.14 ± 0.22 a	1.01 ± 0.11 a	0.08 ± 0.01 a	1.66 ± 1.43 a
10	(*Z*)-Neral	1243	Monoterpene	40.46 ± 0.30 b	64.96 ± 6.69 a	10.56 ± 2.21 c	5.79 ± 1.74 c
11	(*E*)-Nerol	1258	Terpene alcohol	1.65 ± 0.32 ab	2.40 ± 0.73 a	0.72 ± 0.13 ab	0.06 ± 0.01 b
12	Methyl citronellate	1262	Esters	ND ^2^	ND	ND	0.01 ± 0.01
13	Geranial	1272	Monoterpene	118.38 ± 2.69 a	143.62 ± 11.78 a	22.85 ± 5.01 b	12.10 ± 3.74 b
14	Methyl geraniate	1325	Monoterpene	0.05 ± 0.00 b	0.34 ± 0.04 a	0.05 ± 0.01 b	0.04 ± 0.01 b
15	Caryophyllene	1423	Sesquiterpene	0.19 ± 0.01 b	0.83 ± 0.11 a	0.05 ± 0.00 b	ND
16	(*E*)-α-Ionone	1426	Ketones	0.07 ± 0.01 b	0.02 ± 0.00 b	0.17 ± 0.01 a	0.04 ± 0.01 b
17	Nerolidol	1557	Terpene alcohol	0.03 ± 0.00 b	0.08 ± 0.01 a	ND	ND
18	γ-Dodecalactone	1678	Lactones	0.03 ± 0.01 a	0.03 ± 0.01 a	0.01 ± 0.00 a	0.02 ± 0.00 a
		Total		165.95 ± 4.21 a	215.58 ± 34.08 a	35.24 ± 12.63 b	20.56 ± 11.08 b

Values were expressed as mean ± standard error and experiments were conducted with repetitive analyses. An ANOVA test was performed with Duncan’s multiple-range test using the SPSS program and the result was expressed as different letters (a–c). ^1^ RI: retention index; ^2^ ND: not detected.

**Table 5 antioxidants-12-01909-t005:** VOCs’ profiling change (ng/mg) of OBP following the DAS.

No.	Compound	RI ^1^	Classification	DAS			
				30	60	70	80
1	(*Z*)-3-Hexenal	802	Aldehyde	0.04 ± 0.01	ND	ND	ND
2	(*E*)-2-Hexenal	857	Aldehyde	0.45 ± 0.02 a	0.32 ± 0.06 a	0.05 ± 0.01 b	0.07 ± 0.04 b
3	(*E,E*)-2,4-Hexadienal	913	Aldehyde	0.04 ± 0.00	ND	ND	ND
4	1-Octen-3-ol	981	Fatty alcohol	0.06 ± 0.01 b	0.11 ± 0.01 a	0.02 ± 0.00 c	0.04 ± 0.00 bc
5	β-Pinene	983	Monoterpene	ND ^2^	0.12 ± 0.01 a	ND	0.02 ± 0.01 b
6	β-Myrcene	993	Monoterpene	ND	0.13 ± 0.05 a	ND	0.02 ± 0.01 b
7	L-Limonene	1030	Monoterpene	ND	0.14 ± 0.03	ND	ND
8	Eucalyptol	1033	Monoterpene	1.61 ± 0.25 b	5.64 ± 1.34 a	0.09 ± 0.01 b	1.22 ± 0.41 b
9	cis-4-Thujanol	1072	Terpene alcohol	0.11 ± 0.01 a	0.17 ± 0.03 a	ND	ND
10	4-Methyl benzaldehyde	1079	Aldehyde	2.24 ± 0.34 a	ND	0.33 ± 0.04 b	0.32 ± 0.04 b
11	Linalool	1101	Terpene alcohol	4.15 ± 0.46 b	15.30 ± 1.55 a	0.85 ± 0.55 b	15.26 ± 6.71 a
12	(*Z*)-Thujone	1105	Monoterpene	0.11 ± 0.02	ND	ND	ND
13	L-camphor	1146	Monoterpene	0.18 ± 0.03 b	0.40 ± 0.02 a	ND	ND
14	δ-Terpineol	1167	Terpene alcohol	0.24 ± 0.01 a	0.38 ± 0.06 a	0.36 ± 0.18 a	0.14 ± 0.04 a
15	endo-Borneol	1169	Terpene alcohol	ND	ND	0.05 ± 0.01 a	0.03 ± 0.01 ab
16	Terpinen-4-ol	1179	Terpene alcohol	0.05 ± 0.00 b	0.13 ± 0.03 a	0.03 ± 0.01 b	0.04 ± 0.01 b
17	α-Terpineol	1192	Terpene alcohol	1.26 ± 0.07 b	2.24 ± 0.30 a	0.28 ± 0.10 c	0.78 ± 0.23 bc
18	Estragole	1197	Phenylpropene	0.22 ± 0.02 b	3.10 ± 0.22 a	ND	0.26 ± 0.03 b
19	(*Z*)-Neral	1243	Monoterpene	ND	0.58 ± 0.02 a	ND	0.10 ± 0.09 b
20	(*E*)-Neral	1270	Monoterpene	ND	1.04 ± 0.06 a	ND	0.19 ± 0.17 b
21	Estragole	1199	Terpene alcohol	0.07 ± 0.01 b	0.61 ± 0.13 a	ND	0.04 ± 0.02 b
22	Eugenol	1359	Phenylpropene	39.38 ± 8.02 a	54.49 ± 8.44 a	4.42 ± 3.3 b	17.06 ± 2.75 b
23	Methyleugenol	1409	Phenylpropene	165.91 ± 22.84 a	150.64 ± 33.20 a	4.42 ± 2.5 b	5.61 ± 2.56 b
24	Caryophyllene	1423	Sesquiterpene	0.17 ± 0.07 b	1.12 ± 0.52 a	ND	0.17 ± 0.12 b
25	trans-α-Bergamotene	1437	Sesquiterpene	1.96 ± 0.63 b	14.78 ± 3.10 a	0.03 ± 0.01 b	4.56 ± 2.28 b
26	(*E*)-β-Famesene	1459	Sesquiterpene	0.35 ± 0.08 b	2.19 ± 0.61 a	ND	0.17 ± 0.05 b
27	Humulene	1459	Sesquiterpene	0.25 ± 0.12 b	0.73 ± 0.26 a	ND	0.16 ± 0.06 b
28	Methylisoeugenol	1492	Phenylpropene	0.29 ± 0.12 a	0.31 ± 0.02 a	ND	ND
29	Guaiene	1493	Sesquiterpene	ND	0.59 ± 0.14	ND	ND
30	trans-Calamenene	1530	Sesquiterpene	ND	0.15 ± 0.01 a	ND	0.03 ± 0.01 b
31	Nerolidol	1557	Terpene alcohol	0.08 ± 0.02 b	0.18 ± 0.03 a	ND	0.05 ± 0.01 bc
32	T-cadinol	1643	Terpene alcohol	0.89 ± 0.22 b	2.45 ± 0.37 a	0.11 ± 0.03 c	1.49 ± 0.37 b
		Total		221.6 ± 17.2 b	259.0 ± 14.9 a	11.1 ± 3.8 d	48.5 ± 7.3 c

Values were expressed as mean ± standard error and experiments were conducted with repetitive analyses. An ANOVA test was performed with Duncan’s multiple-range test using the SPSS program and the result was expressed as different letters (a–c). ^1^ RI: retention index; ^2^ ND: not detected.

**Table 6 antioxidants-12-01909-t006:** VOCs’ profiling change (ng/mg) of SO following the DAS.

No.	Compound	RI ^1^	Classification	DAS			
				30	60	70	80
1	(*E*)-2-Hexenal	857	Aldehyde	0.61 ± 0.17 a	0.50 ± 0.18 a	0.11 ± 0.02 b	0.14 ± 0.05 b
2	α-Pinene	939	Monoterpene	0.66 ± 0.05 a	0.25 ± 0.19 b	0.11 ± 0.07 b	0.11 ± 0.08 b
3	Camphene	953	Monoterpene	0.51 ± 0.11 a	0.31 ± 0.17 b	0.14 ± 0.07 b	0.09 ± 0.07 b
4	1-Octen-3-ol	981	Fatty alcohol	0.06 ± 0.01 ab	0.08 ± 0.01 a	0.03 ± 0.00 b	0.04 ± 0.02 b
5	β-Pinene	983	Monoterpene	0.59 ± 0.12 a	0.48 ± 0.23 ab	0.17 ± 0.11 b	0.22 ± 0.12 ab
6	β-Myrcene	993	Monoterpene	0.49 ± 0.01 a	0.06 ± 0.02 b	0.06 ± 0.08 b	0.11 ± 0.07 b
7	α-Phellandrene	1007	Monoterpene	0.02 ± 0.00 a	0.03 ± 0.00 a	0.03 ± 0.02 a	0.03 ± 0.04 a
8	(*E,E*)-2,4-Heptadienal	1016	Aldehyde	0.01 ± 0.00 a	0.02 ± 0.00 a	0.02 ± 0.00 a	0.03 ± 0.01 a
9	α-Terpinene	1019	Monoterpene	0.05 ± 0.00 a	0.05 ± 0.00 a	ND ^2^	ND
10	o-Cymene	1025	Monoterpene	0.05 ± 0.01 a	0.04 ± 0.01 ab	0.02 ± 0.01 b	0.03 ± 0.01 b
11	Limonene	1030	Monoterpene	0.71 ± 0.09 a	0.13 ± 0.05 a	0.09 ± 0.08 a	0.68 ± 0.61 a
12	Eucalyptol	1033	Monoterpene	7.40 ± 0.53 a	5.87 ± 0.79 ab	2.36 ± 0.33 c	3.84 ± 2.15 bc
13	cis-Sabinene hydrate	1076	Monoterpene	0.16 ± 0.01 a	0.18 ± 0.01 a	0.06 ± 0.01 b	0.08 ± 0.03 b
14	4-Methyl benzaldehyde	1079	Aldehyde	0.19 ± 0.07 c	1.49 ± 0.09 a	0.67 ± 0.18 b	0.64 ± 0.30 bc
15	α-Terpinolene	1089	Monoterpene	0.43 ± 0.05 a	0.31 ± 0.03 b	0.13 ± 0.01 c	0.14 ± 0.03 c
16	Linalool	1101	Terpene alcohol	0.50 ± 0.12 b	1.18 ± 0.29 a	0.44 ± 0.07 b	0.46 ± 0.19 b
17	(*Z*)-Thujone	1105	Monoterpene	30.83 ± 4.68 ab	36.53 ± 3.38 a	22.81 ± 1.28 b	32.72 ± 5.40 ab
18	(*E*)-Thujone	1116	Monoterpene	7.60± 1.71 a	4.57 ± 1.06 b	3.09 ± 0.26 b	4.06 ± 0.72 b
19	(+)-Sabinol	1145	Terpene alcohol	0.26 ± 0.06 a	0.23 ± 0.06 a	0.13 ± 0.03 a	0.27 ± 0.08 a
20	L-camphor	1146	Monoterpene	24.55 ± 2.14 b	37.77 ± 4.63 a	9.63 ± 3.37 c	11.79 ± 1.89 c
21	Isothujol	1168	Terpene alcohol	0.52 ± 0.24 a	0.74 ± 0.24 a	2.47 ± 3.07 a	0.33 ± 0.04 a
22	endo-Borneol	1169	Terpene alcohol	0.94 ± 0.12 ab	2.11 ± 1.22 a	0.48 ± 0.10 b	0.45 ± 0.06 b
23	Terpinen-4-ol	1179	Terpene alcohol	0.34 ± 0.09 b	0.59 ± 0.07 a	0.33 ± 0.10 b	0.29 ± 0.11 b
24	p-Cymen-8-ol	1185	Terpene alcohol	0.10 ± 0.01 ab	0.11 ± 0.04 a	0.04 ± 0.00 b	0.04 ± 0.00 b
25	α-Terpineol	1192	Terpene alcohol	0.40 ± 0.04 b	0.64 ± 0.16 a	0.20 ± 0.06 b	0.26 ± 0.06 b
26	Estragole	1198	Phenylpropene	0.66 ± 0.10 b	3.24 ± 1.13 a	1.06 ± 0.36 b	0.58 ± 0.11 b
27	(*E*)-Carveol	1219	Terpene alcohol	0.15 ± 0.01 ab	0.18 ± 0.07 a	0.04 ± 0.02 b	0.07 ± 0.05 ab
28	(*Z*)-Neral	1243	Monoterpene	0.23 ± 0.03 a	0.18 ± 0.01 ab	0.10 ± 0.07 bc	0.03 ± 0.01 c
29	(*E*)-Neral	1270	Monoterpene	0.52 ± 0.07 a	0.40 ± 0.04 a	0.19 ± 0.11 b	0.03 ± 0.00 b
30	(−)-Bornyl acetate	1286	Terpene alcohol	2.11 ± 0.70 a	2.70 ± 0.57 a	0.60 ± 0.17 b	0.43 ± 0.10 b
31	Methyleugenol	1409	Phenylpropene	ND	0.36 ± 0.08 a	0.11 ± 0.03 b	0.06 ± 0.03 b
32	Caryophyllene	1423	Sesquiterpene	1.18 ± 0.10 a	1.86 ± 0.42 a	1.48 ± 0.28 a	1.41 ± 0.16 a
33	Humulene	1459	Sesquiterpene	8.34 ± 1.07 a	16.13 ± 6.26 a	7.83 ± 0.63 a	10.43 ± 0.91 a
34	(+)-Viridiflorol	1595	Terpene alcohol	3.26 ± 0.43 b	9.15 ± 0.62 a	6.48 ± 2.01 ab	6.00 ± 1.34 ab
		Total		94.45 ± 7.54 a	128.47 ± 8.49 a	61.51 ± 3.38 b	75.89 ± 12.75 b

Values were expressed as mean ± standard error and experiments were conducted with repetitive analyses. An ANOVA test was performed with Duncan’s multiple-range test using the SPSS program and the result was expressed as different letters (a–c). ^1^ RI: retention index; ^2^ ND: not detected.

## Data Availability

The data presented in this study are available in the article and in Appendix A.

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
