# Peer review of "Effects of Harvest Timing on Phytochemical Composition in Lamiaceae Plants under an Environment-Controlled System"

_antioxidants, 2023, doi:10.3390/antiox12111909_

Round 1

Reviewer 1 Report

Please see in the attachment.

Reviewer 2 Report

Please find the comments in the attached file

Reviewer 3 Report

The paper is well written and easy to understand, although the authors did not avoid typos. The text should be carefully checked in this regard.

I can recommend the manuscript for publication after minor revision:

Detailed comments:

1.       Line 95: remove "In" from the formula; digits should be in subscripts

2.       Line 110: 1N concentration, please provide in mol/L

3.       Line 115: in the topic of section 2.4. you should provide the name of the compound rather than its abbreviation (RA) or provide the name and abbreviation; similarly in line 137 (RAC)

4.       In line 155: in the formula for calculating DPPH radical scavenging activity is: Abs sampe – must be Abs sample; similarly in line 166 for ABTS formula;

5.       In line 156 and 167 is: of sample solution, maybe it would be better: of tested sample

6.       No description for Figure 2(C). Figure 2(C) is not very visible and should be corrected.

Round 2

Reviewer 1 Report

The quality of the manuscript has been highly improved with the exception of still lacking List of Abbreviations. However, this requirement should be expressed by the Editors, so I leave this point to their decision.

The quality of English has been improved compared to the former version.

Author Response

Following the editor's comments, it was modified to improve the English language.
